# Development and Validation of a Mucosal Antibody (IgA) Test to Identify Persistent Infection with Foot-and-Mouth Disease Virus

**DOI:** 10.3390/v13050814

**Published:** 2021-05-01

**Authors:** Jitendra K. Biswal, Antonello Di Nardo, Geraldine Taylor, David J. Paton, Satya Parida

**Affiliations:** The Pirbright Institute, Ash Road, Pirbright, Surrey GU24 0NF, UK; jkubiswal@gmail.com (J.K.B.); antonello.dinardo@pirbright.ac.uk (A.D.N.); geraldine.taylor@pirbright.ac.uk (G.T.); david.paton@pirbright.ac.uk (D.J.P.)

**Keywords:** foot-and-mouth disease, ELISA, mucosal IgA, carrier (persistent infection), post-outbreak surveillance, nasal, saliva and oro-pharyngeal fluid (OPF)

## Abstract

It is well known that approximately 50% of cattle infected with foot-and-mouth disease (FMD) virus (FMDV) may become asymptomatic carrier (persistently infected) animals. Although transmission of FMDV from carrier cattle to naïve cattle has not been demonstrated experimentally, circumstantial evidence from field studies has linked FMDV-carrier cattle to cause subsequent outbreaks. Therefore, the asymptomatic carrier state complicates the control and eradication of FMD. Current serological diagnosis using tests for antibodies to the viral non-structural proteins (NSP-ELISA) are not sensitive enough to detect all carrier animals, if persistently infected after vaccination and do not distinguish between carriers and non-carriers. The specificity of the NSP ELISA may also be reduced after vaccination, in particular after multiple vaccination. FMDV-specific mucosal antibodies (IgA) are not produced in vaccinated cattle but are elevated transiently during the acute phase of infection and can be detected at a high level in cattle persistently infected with FMDV, irrespective of their vaccination status. Therefore, detection of IgA by ELISA may be considered a diagnostic alternative to RT-PCR for assessing FMDV persistent infection in ruminants in both vaccinated and unvaccinated infected populations. This study reports on the development and validation of a new mucosal IgA ELISA for the detection of carrier animals using nasal, saliva, and oro-pharyngeal fluid (OPF) samples. The diagnostic performance of the IgA ELISA using nasal samples from experimentally vaccinated and infected cattle demonstrated a high level of specificity (99%) and an improved level of sensitivity (76.5%). Furthermore, the detection of carrier animals reached 96.9% when parallel testing of samples was carried out using both the IgA-ELISA and NSP-ELISA.

## 1. Introduction

Foot-and-mouth disease (FMD) is a highly contagious vesicular disease caused by FMD virus (FMDV), an aphthovirus within the family *picornaviridae* that infects both domesticated and wild cloven-hoofed animals [1]. The classical FMD symptoms in ruminants are characterised by fever, inappetence, lameness, excess salivation and vesicles in and around the mouth, teats and feet. These clinical signs normally subside approximately 10–14 days post-infection [2]. However, up to 50% of FMD-recovered cattle may harbour virus in their oro-pharyngeal and naso-pharyngeal cavity at 28 or more days post-infection and are known as FMDV-carriers or persistently infected animals [3,4]. This asymptomatic carrier state of FMD complicates the control and eradication of the disease. The duration of the FMDV-carrier state may be influenced by a combination of viral and host-factors, and can last from months to years [5]. Although transmission of FMD virus from domestic animal carriers to susceptible naïve animals has not been demonstrated under experimental conditions [6,7], circumstantial evidence from field studies has linked FMDV-carrier cattle to subsequent outbreaks [8,9,10]. Furthermore, a recent experimental study has demonstrated clinical infection in naïve cattle when inoculated with oro-pharyngeal fluids (OPF) obtained from carrier animals [11]. Since FMDV-carriers may be considered a risk for transmitting infection, they must be identified by post-vaccination serosurveillance to substantiate freedom from infection to regain the “FMD-free status without vaccination” for the purpose of international trade [12,13]. Carrier animals persistently infected with FMDV can be identified by detection of the virus in OPF collected with a probang sampling cup. However, recovery of infectious virus or viral genome from such oro-pharyngeal scrapings of FMDV persistently infected cattle is intermittent [12,14].

Tests for the detection of antibodies to FMDV non-structural proteins (NSP) have been used for detection of infection in vaccinated animals (DIVA). However, the currently validated NSP antibody tests [15] may not detect all infected animals within a vaccinated population [16] and do not distinguish between carriers and those that have eliminated FMDV. Therefore, new NSP tests or alternative tests which can be used either as screening tests or confirmatory tests to the existing NSP tests are needed.

Several studies describe the presence of FMDV-specific mucosal IgA (IgA) in oro-pharyngeal fluid as an indicator of FMDV persistence [14,17,18,19]. An IgA-ELISA, using saliva samples, to detect FMD carrier cattle following vaccination and challenge exposure has been developed previously [14]. Although this IgA-ELISA can detect carrier cattle in vaccinated and unvaccinated populations, the non-specific reactions with some saliva samples from uninfected animals have hampered the introduction of the test (unpublished results).

The above findings led us to develop and validate a new mucosal IgA assay, using saliva, nasal and OPF samples, as a confirmatory or screening test for detection of persistently FMDV infected cattle.

## 2. Materials and Methods

### 2.1. Animals and Sample Collection

#### 2.1.1. Single Dose O Manisa Vaccination and O UKG Challenge Experiments

Saliva, nasal and OPF samples were collected from four vaccine challenge experiments, each consisting of 25 Holstein-Friesian cattle, aged 4–8 months, carried out in biosecurity containment at the Pirbright Institute, Pirbright, U.K. Experiments had been designed to evaluate vaccine induced protection against challenge 21 or 10 days later, by contact with animals infected with a semi-heterologous FMDV strain. Animals in each experiment were assigned with 2 letter identifiers (UV, UY, VD/VE, and VH) coupled with animal-specific numbers. In each experiment, 20 cattle were vaccinated with oil adjuvanted FMDV type O Manisa emergency vaccine obtained from the UK FMDV antigen reserve, and compliant with the Office International des Epizooties (OIE) guidelines for freedom from non-structural proteins (NSPs). In each experiment five cattle were included as unvaccinated control. In UV and VH animal experiments, the vaccine used represented a formulation previously determined to have a potency of 18PD_50_ (according to the European pharmacopoeia cattle potency test), whereas in the UY, VD/VE animal experiments, a 10 times increase per bovine dose of antigen was included in the vaccine formulation. The challenge was performed by 5 days of contact with donor cattle that had been infected with O UKG 34/2001 by tongue inoculation. The experimental details have been extensively reported in earlier publications [14,20,21,22,23] and are shown in the Appendix A. 

#### 2.1.2. Repeat Dose O Manisa Vaccination and O UKG Challenge Experiment

In order to evaluate the effect of multiple vaccinations on anti-FMDV mucosal antibody titres, mucosal fluids (saliva, nasal, probang) were collected from cattle vaccinated with O Manisa vaccine (18PD_50_) three times at 21-day intervals and then challenged with O UKG 34/2001 FMD virus on the 35th day after the 3rd vaccination. This challenge was performed by contact with the donor animals from the above-mentioned UY experiment starting six days after FMDV tongue inoculation. 

#### 2.1.3. Naïve Cattle

To determine the specificity and cut-off value of the mucosal IgA test, saliva fluid samples (*n* = 875), were collected from cattle that had had no contact with FMDV. These cattle comprised animals from the Mayfield dairy farm, Compton, UK; from the Republic of Ireland; from animals sourced from the UK, at the beginning of vaccine challenge/vaccine potency experiments at the Pirbright Institute, before any administration of vaccine or virus. Similarly, nasal fluids (*n* = 224) and OPF samples (*n* = 188) had been collected from individual naïve cattle before use in the different vaccine challenge/vaccine potency experiments and were used for the determination of cut-off values in nasal and probang IgA tests, respectively. Saliva samples were also collected from these animals for the cut-off estimation of the saliva test.

#### 2.1.4. Sample Collection

Saliva and nasal fluids were collected from the vestibule and underneath the tongue of the mouth and from the nostril, respectively by using a 1/6th portion of regular size cotton tampons (Tampax^®^, Hungary) or Sarsted saliva collection kits^®^ pre-dampened by the addition of 0.5 mL of phosphate buffered saline (pH 7.5). The tampon swab was held by forceps to collect the samples. In the laboratory, approximately 1–2 mL of saliva/nasal fluid samples were extracted from each tampon/saliva collection kit by compression within the barrel of a syringe or by centrifugation for 10 min at 1862× *g* before storage at −20 °C [14]. Samples from tampons and kits were compared for volume and presence of antibody and were found equivalent. OPF were collected by using a probang sampling cup [24]. Blood samples were also collected for detection of anti-NSP antibody in serum using Vacutainers^®^ (BD, USA). 

### 2.2. Test Procedures

#### 2.2.1. Virological Tests

Results of virus isolation (VI) and RT-PCR from probang fluids that had already been documented [14,20,21,22,23], were considered as the gold standard for the detection of carrier cattle. Probang sample positive animals, either by RT-PCR or by VI or by both techniques after 28 days post-challenge, were considered as carrier animals. As shown in previous studies [1,3], perhaps due to variability in the material collected by probang sampling, intermittently positive virological results may be obtained in both the tests. However, the animals were considered as carriers if the results were positive on at least one occasion on or after 28 days post-challenge. The same approach was used to characterise the samples and animals from the experimental cattle vaccinated three times before challenge that have not been previously reported.

#### 2.2.2. Serological Tests

Pre-documented results [14,20,21,22,23] of the PrioCHECK^®^-NSP assay (Prionics AG) have been used to estimate the concordance between 3ABC-NSP test and IgA assay for the detection of carriers. ThePrioCHECK^®^-NSP assay was also used to measure anti-NSP antibodies in samples from the cattle vaccinated three times prior to challenge. The PrioCHECK^®^-NSP is a competitive blocking ELISA that measures the competition between serum anti-FMDV NSP antibody and a NSP-3B specific monoclonal antibody for binding to a recombinant 3ABC NSP of FMDV [25,26]. The assay was conducted as per the manufacturer’s instructions.

#### 2.2.3. Indirect Sandwich ELISA for Detection of IgA Antibodies

An indirect ELISA for the detection of IgA antibodies to structural proteins has been developed previously [14]. However, to increase the specificity of the salivary IgA test, two separate negative antigens (FMDV SAT2 and BHK-21 cell lysate) and a blocking buffer were included as negative controls in the ELISA plate. The final OD was calculated after deducting the OD value of the negative control from the OD value of the specific antigen. The best negative control selected in the salivary IgA assay has been used subsequently in the nasal and OPF IgA ELISA. 

Odd-numbered columns of the ELISA plates (Nunc, Roskilde, Denmark) were coated with 0.1 M carbonate/bicarbonate buffer, pH 8.0–8.4 (50 μL/well) containing rabbit anti-FMDV antiserum (O Manisa), while the even-numbered columns were coated with SAT2 rabbit anti-FMDV antiserum. In addition to the SAT2 control, in the case of the salivary IgA test, BHK-21 cell lysate and blocking buffer control wells were coated with rabbit anti-FMDV antiserum (O Manisa). After incubating the coated plate at 4 °C overnight, the respective pre-titrated antigens diluted in blocking buffer were added. Following washing, test samples (7 μL of saliva/probang or 2 μL of nasal fluid) along with in-house test positive and negative standards (controls) were added to the blocking buffer (43/48 μL) in the plates and incubated for 1 h at 37 °C. After a further wash, specific bovine IgA was detected using a polyclonal rabbit anti-bovine IgA HRPO conjugate. Plates were finally washed three times and the test was developed by the addition of substrate. The reaction was stopped after 10 min by addition of 1 M sulphuric acid and the optical density (OD) from plates were read on a multi-channel spectrophotometer (Molecular Devices, Inc., San Jose, CA, USA) at 490 nm (A_490_).

The test results were expressed in terms of percentage of positivity (PP) which was determined as follows:$$OD_{corrected}=OD_{sample}-OD_{control}$$
$$PP=\frac{100\times OD_{corrected\left(test\right)}}{OD_{corrected\left(positive\;control\right)}}$$

### 2.3. Statistical Analysis

Comparison of the anti-FMDV IgA response between the carrier and non-carrier groups was performed using the student’s t-test, whereas analysis of variance (ANOVA) was used to assess differences in the anti-FMDV IgA antibody level estimated from the nasal, saliva and oro-pharyngeal fluids. To determine the performance of the IgA-ELISA, nonparametric estimations of receiver operating characteristic (ROC) curves were run in Stata 14 SE (Stata Corp, LP), evaluating the diagnostic performance (sensitivity (Se) and specificity (Sp)) at different cut-off points. The maximum-likelihood ROC model was used to estimate the area under the curve (AUC) with associated 95% confidence intervals. A Bayesian model was further parametrised for assessing the sensitivity and specificity of the IgA-ELISA using a probabilistic constraint framework) [27]. The model was computed using WinBUGS 1.4.3 (MRC Biostatistics unit, Cambridge, UK) [28]. The MCMC chain was run for 50,000 with burn-in set at 10% of the chain.

### 2.4. Ethics Statement

All calf experiments were performed under the regulations of the Home Office Scientific Procedures Act (1986) of the United Kingdom, and had been approved by The Pirbright Institute Animal Welfare & Ethical Review Body.

## 3. Results

### 3.1. Detection of FMDV Carrier Cattle by VI and RT-PCR

Out of 80 vaccinated and 20 unvaccinated control cattle from four vaccine-challenge experiments involving a single dose of vaccine, 32 and 5 cattle were detected as FMDV carriers, respectively. The carrier status was identified by combining the VI and real-time RT-PCR tests (VI + RT-PCR). Analyses of OPF samples for detection of carriers in UV, UY, VD/VE and VH have been previously published [12,14,21]. All cattle that received multiple vaccine doses, were clinically protected against FMDV after challenge. Indeed, no virus/viral genome was identified by VI and RT-PCR from the OPF samples collected from the multiple vaccinated-challenged cattle.

### 3.2. Development and Validation of IgA ELISA for the Evaluation of Anti-FMDV IgA Antibody Response in the Mucosal Fluids

#### 3.2.1. Selection of Suitable Negative Antigen Control to Increase the Sensitivity and Specificity of the Salivary IgA Assay

A test of equality of ROC areas was done using saliva samples of known uninfected and infected cattle tested using IgA assays constructed with the three different antigen controls (i.e., FMDV SAT2 and BHK-21 cell lysate, and blocking buffer). IgA ELISA data on saliva samples from naive uninfected cattle and from vaccinated uninfected cattle were used separately (Table 1 and Table 2). ROC curves were plotted using the IgA-ELISA data originated from 32 cattle during the vaccination period (vaccinated uninfected cattle saliva, *n* = 78) and then after they were infected and became persistently infected (vaccinated persistently infected cattle saliva samples, *n* = 300), the samples from carrier cattle were collected at weekly intervals between 28 and 168 days post challenge (dpc). Although the AUC (0.93) estimated for both the BHK-21 and SAT2 negative antigen controls was relatively larger than that of the blocking buffer (BB) control (AUC = 0.909) (Figure 1), the sensitivity for the detection of FMDV carrier by IgA-ELISA was found to be highest using the SAT2 antigen control (Table 1). Therefore, considering the estimated sensitivity at different cut-off points (Table 1), the SAT2 antigen control was selected as the most suitable negative antigen control for the serotype O specific IgA ELISA.

Similarly, ROC curves were also plotted using the IgA-ELISA data from saliva samples collected from 875 individual naïve cattle and from saliva samples (*n* = 300) collected from 32 vaccinated persistently infected cattle collected at weekly intervals between 28 and 168 dpc (Figure 2). The AUC for the BHK-21 antigen control was found to be the highest among the three antigen controls (Figure 2), but the sensitivity for the detection of FMDV carrier animals by IgA-ELISA was again found to be highest using the SAT-2 antigen control (Table 2). Therefore, the SAT2 control was selected as the suitable negative antigen control for subsequent analysis and validation of IgA-ELISA using saliva, nasal and OPF.

#### 3.2.2. Comparison of Diagnostic Sensitivity and Diagnostic Specificity for Anti-FMDV IgA-ELISAs Using Saliva, Nasal and Oro-Pharyngeal Fluid Samples

Diagnostic characteristics of the salivary IgA ELISA at different cut-off points were estimated by the non-parametric ROC analysis method (Table 1 and Table 2). Considering the Se (sensitivity) and Sp (specificity), a suitable cut-off value of 35 percentage of positivity (PP) was selected for the salivary IgA-ELISA. At 35 PP a Se of 68.92% and Sp of 97.60% was obtained for the salivary IgA ELISA (Table 2). For the determination of the diagnostic parameters of the nasal IgA-ELISA, nasal fluids from individual naïve cattle (*n* = 224) and carrier cattle (300 samples from 32 carrier animals) collected at weekly intervals between 28 and 168 dpc were used. For the nasal IgA-ELISA, a cut-off value of 35 PP was selected with a Se of 76.53% and a Sp of 99.11% (Table 3). Similarly, 188 OPF samples from individual naïve cattle and 276 OPF samples from carrier cattle (*n* = 37) collected at weekly intervals between 28 and 161 dpc, were analysed by the IgA-ELISA assay and a suitable cut-off value of 35 PP was reported (Table 4). At the 35 PP, a Se of 59.35% and a Sp of 99.47% was obtained (Table 4).

Since nasal sample-based IgA-ELISA performed better as compared to the saliva and OPF samples, the diagnostic Sp and Se of the nasal IgA-ELISA was also calculated using a Bayesian framework with probabilistic constraints. The Bayesian model provides a framework with multiple iterations to assess uncertainty in order to improve the accuracy of the ROC calculations. By using the Bayesian model, a specificity of 99.3% and a sensitivity of 85.4% was obtained for the nasal IgA-ELISA (Table 5).

#### 3.2.3. Comparative Anti-FMDV IgA Response in the Mucosal Fluid (Saliva, Nasal and Probang) Samples of Carrier and Non-Carrier Cattle

In all the mucosal fluids, the mean anti-FMDV IgA titre remained below the cut-off value (35 PP) during the entire vaccination period (Figure 3). However, in the vaccinated carrier cattle but not in non-carriers, the mean IgA response in nasal fluid started to rise within 7–14 days post-challenge (dpc). In comparison, the IgA response both in saliva and probang samples was delayed up to 14–21 dpc (Figure 3). By 21 dpc, the nasal IgA response remained above the cut-off value whereas both the saliva and OPF IgA responses were seen to be positive only after 28 dpc. The mean anti-FMDV IgA response in mucosal fluids of all the 5 unvaccinated carrier cattle was found to be above the cut-off value only after 35 dpc (Figure 4).

The mean anti-FMDV IgA responses in all the mucosal fluids of vaccinated and unvaccinated carrier animals were seen to be significantly higher (*p* ≤ 0.05) than that of the respective non-carrier animals (Figure 3 and Figure 4). In vaccinated carriers, a higher mean anti-FMDV IgA antibody response was observed in the nasal samples than the saliva and probang samples during many of the sampling days beyond 28 days post-challenge (dpc) (Figure 3). However, a statistically significant difference (*p* ≤ 0.05) in the anti-FMDV IgA response was found between the three fluids only on 35 dpc (*p* = 0.03), 42 dpc (*p* = 0.001) and 63 dpc (*p* = 0.04).

#### 3.2.4. Analysis of Sensitivity Concordance between Nasal IgA-ELISA and PrioCHECK-NSP Test for the Detection of FMDV Carrier Cattle

As the FMDV NSP antibody assay has been used for the detection of FMDV carrier cattle, the level of concordance between the commercially validated PrioCHECK-NSP test and the nasal IgA-ELISA for the detection of FMDV carriers was determined. Out of 32 vaccinated-challenged carrier cattle detected by VI and/or RT-PCR, a total of 29 cattle were scored positive either by the nasal IgA assay or by the PrioCHECK-NSP test (Table 6). However, from these 29 detected carrier cattle, only 27 cattle were found to be concordantly positive by both IgA and NSP antibody tests. The PrioCHECK-NSP assay scored another 18 vaccinated cattle as seropositive after challenge, in addition to the 29 cattle that were virologically confirmed carriers (Table 6). Combining the results from the nasal IgA and the NSP antibody test, 31 out of 32 vaccinated carriers were scored positive by one or both tests, resulting in an enhanced sensitivity of 96.87%.

Multiple samples were available from many of the vaccinated and challenged cattle and, in some cases, the results changed over time so that both virological and serological tests sometimes failed to detect carriers at one or more timepoints. Therefore, it was decided to calculate the sensitivity concordance by using the data originating from individual sampling days on and after 28 days post challenge. Considering the number of known carrier cattle on different days of sampling period as 100%, the sensitivity of all three assays were calculated (Table 7). The sensitivity of the combined VI + RT-PCR assay for the detection of carriers varied from 0 to 78% (Table 6). The sensitivity of IgA-ELISA for detection of FMDV persistent cattle varied from 62 to 88%, while the performance of PrioCHECK-NSP test for the detection of carriers varied from 75 to 90% (Table 7). By combining both the IgA assay and NSP test, 75–100% of known carriers were detected by one or both tests (Table 7).

#### 3.2.5. Anti-FMDV IgA and Anti-NSP Antibody Responses in Cattle Challenged after Repeated Vaccination

Repeatedly vaccinated cattle (*n* = 6) were clinically protected and none of them scored as carriers after 28 dpc by VI or RT-PCR tests (data not shown). One animal (VC14) scored positive for anti-FMDV NSP antibody by PrioCHECK-NSP assay after the third vaccination (56 dpv), while animal VC19 scored positive, just above the threshold cut-off, on 63 dpv and 70 dpv. However, these two cattle became seropositive after challenge (Figure 5). FMDV specific IgA antibody responses in the nasal samples of the cattle remained below the threshold (35 PP) value at all times prior to challenge (Figure 5). However, a rise in IgA antibody was found in VC15 immediately after the second vaccination (Figure 5), albeit remaining below the threshold value. After two to three weeks of challenge, only one animal (VC 14) scored positive in the IgA antibody ELISA.

## 4. Discussion

Since FMDV carriers may be considered to pose a risk of passing on infection, to regain the FMD-free status without ongoing vaccination for the purpose of international trade, with or without the use of temporary emergency vaccination, post-outbreak surveillance is required to demonstrate the absence of infection [29]. Virus isolation and RT-PCR are two established methods to detect virus or viral genome from oro-pharyngeal fluids collected using a metallic probang cup [30]. Virus isolation is as sensitive as RT-PCR during the early phase of infection, whereas RT-PCR detects more carriers in the later phase of infection in both cattle and sheep [12,16]. Even after combining both virus isolation and RT-PCR test results, at all the time points of sampling 100% carriers are not detected from the OPF samples collected sequentially in time [12], which is a major constraint in FMD surveillance. Besides this, the approach is cumbersome for mass screening after widespread use of emergency vaccines. Therefore, there is a need for alternative or supplementary methods to detect subclinical infection with FMDV in vaccinated herds.

The presence of anti-FMDV IgA antibody in saliva has been described as an indicator of oro-pharyngeal replication of FMDV [14]. Though an indirect IgA-ELISA using saliva samples to detect FMDV carrier cattle following vaccination and challenge exposure has been developed earlier, the assay was unsatisfactory on its diagnostic parameters. Therefore, an attempt has been made to develop and validate a modified IgA-ELISA targeting an increased specificity. Non-specificity in the assay may be either due to the high content of detached cells, proteases and tissue particles in the saliva samples [17], or due to the use of tissue culture derived inactivated crude antigen. In order to achieve a better diagnostic specificity, different negative antigen controls (heterologous SAT2 antigen, BHK-21 cell lysate and blocking buffer without any antigen) were evaluated in the newly developed salivary IgA-ELISA. Accordingly, the best format of the IgA-ELISA was selected with respect to both specificity and sensitivity using receiver-operating characteristic (ROC) curve analysis [31].

Normally, naïve samples are used to estimate the specificity of the diagnostic assay. However, considering the fact that serosurveillence should normally be conducted after emergency vaccination, it was decided to include saliva samples from both naive and vaccinated uninfected animals as a negative population control in ROC analysis for the selection of suitable negative antigen control.

The anti-FMDV IgA response was found to be higher in nasal fluids of FMDV vaccinated and sub-clinically infected carrier animals than saliva and probang fluids. However, both in the acute (for unvaccinated control animals) and in the persistent infection phase (both vaccinated and unvaccinated), the IgA levels were seen to be increased in all three mucosal fluids (saliva, nasal and probang) compared to samples from the vaccinated uninfected, vaccinated recovered and unvaccinated recovered animals. After 28 days post-challenge, the anti-FMDV IgA responses in the nasal and saliva fluids of both vaccinated and unvaccinated carrier animals were found to be significantly higher than those from the non-carriers. The high level of anti-FMDV IgA in the mucosal fluids may be due to the constant stimulation of the local mucosal immune system by persistent FMDV on the dorsal soft palate or naso-pharynx. Furthermore, the presence of IgA antibody on the nasal mucosal surface is a genuine FMD specific antibody response sustained by resident antibody secreting B-cells, rather than plasma transudation [17].

From the ROC estimations of diagnostic performance, a cut-off value of 35 percentage of positivity (35 PP) was determined for IgA-ELISA using the saliva, nasal and OPF. It was found that nasal IgA-ELISA performed better compared to saliva and OPF IgA-ELISA. With the cut-off value of 35 PP a specificity of 99.11% and sensitivity of 76.53% was obtained for nasal IgA-ELISA by ROC analysis, whilst these parameters were found to increase after Bayesian model fitting (Sp = 99.3% and Se = 85.4%). As this test has similar sensitivity and specificity to the PrioCHECK-NSP test, the IgA assay can be used as screening test for the detection of carrier cattle. The international workshop on serological testing for FMD previously organised in Brescia, Italy, recommended the use of more than one NSP test to increase the overall diagnostic specificity [13]. In this case, the second confirmatory test should be selected with at least equal specificity and a good sensitivity. As the IgA test detects mucosal antibody and the PrioCHECK-NSP test detects anti-FMDV humoral antibody, using one of the tests for screening and the other as confirmatory test may help to increase the specificity and sensitivity to detect the FMDV carriers. The third option is to run both tests in parallel. In the present study, the concordance of sensitivity between the PrioCHECK-NSP and nasal IgA test was found to be 84.37%. By combining both the nasal-IgA test and PrioCHECK-NSP test in parallel the overall test sensitivity was increased to 96.87%.

Considering that there are no tests with perfect sensitivity and specificity, no sero-surveillance can provide an absolute guarantee of freedom from infection. This applies even to a situation when the whole population has to be tested, e.g., the vaccinated cattle population in an EU country after emergency vaccination. Therefore, serosurveillance should be seen as part of a package of risk mitigation measures that will include movement restrictions, epidemiological tracing and clinical surveillance [13]. The problem of imperfect test sensitivity can be mitigated to some extent by attributing a status to a herd or population and then applying appropriate disease control measures. A herd could be classified as infected after one animal has reacted positively. However, as no test has a specificity of 100%, many large herds may have to be considered infected due to false positive test results. As this is unacceptable, simultaneous testing by using serological methods with a very high specificity would be needed [13], albeit this would have the detrimental effect of decreasing the sensitivity. An alternative approach would be the culling of any animal reacting positive without classifying the herd as infected in the absence of further evidence of infection, e.g., clustering or increasing numbers of reactors or epidemiological information. In this case, the requirements for specificity could be lower and a higher sensitivity may be achievable [13].

During this study, the PrioCHECK NSP assay also scored positive for 18 cattle that were confirmed as non-carrier by VI and RT-PCR testing done at weekly intervals. This result confirms that NSP tests are not specific for the detection of carriers, as NSP antibody responses occur in infected animals that go on to eliminate FMDV. In addition, the NSP test may not be useful for individual animal screening, as cattle that have been vaccinated with high potency vaccine may produce stronger and earlier neutralising antibodies which limit the viral replication and sometimes fail to develop antibodies to NSPs following infection, but these animal could potentially become carriers [30,32]. Further in endemic countries where vaccine is not purified from NSP and multiple vaccination is practiced, there is an increased chance of NSP seroconversion without infection.

Although inactivated FMD vaccine, when administered parenterally, stimulates very little or no FMDV specific mucosal immune response [14,33], an immediate question arises about the effect of repeated vaccination on the mucosal anti-FMDV IgA response, particularly in the endemic countries where bi-annual prophylactic vaccination is carried out. In order to address this question, the opportunity was taken to test nasal and saliva samples for the detection of anti-FMDV specific antibody in repeatedly vaccinated animals (three times emergency vaccination at 21-day intervals). Anti-FMDV IgA antibody responses in nasal and saliva samples of six cattle remained below the cut-off value after multiple vaccinations, although a sub-threshold peak in IgA response was found for one animal, VC15, after the second vaccination. The anti-FMDV IgA responses in these repeatedly vaccinated animals suggest that the IgA assay may not be subject to non-specificity from multiple vaccinations even in endemic countries, although more animals need to be tested to confirm this. Furthermore, the analysis of serum samples originating from repeatedly vaccinated animals showed an increased humoral immune response of the anti-FMDV NSP antibody following the third vaccination in animals VC14 and VC19. This result supported the earlier finding [34] where the authors suggested that if animals received multiple doses of FMD vaccine, a small proportion of them might test positive without actual exposure to wild type FMD virus. The detection of NSP antibody in repeatedly vaccinated animals prior to challenge may be due to the presence of trace amounts of contaminating NSP proteins in the commercial FMD vaccine.

In conclusion, from the analysis of anti-FMDV IgA response in mucosal fluids, it is evident that levels of FMDV specific IgA become elevated transiently during the acute phase of infection and were stronger in FMDV carrier animals irrespective of vaccination status. Out of three mucosal fluids, nasal secretion contained the highest level of anti-FMDV IgA antibody. However, a statistically significant difference (*p* ≤ 0.05) in the anti-FMDV IgA antibody response was found between the three fluids only on 35 dpc (*p* = 0.03), 42 dpc (0.001) and 63 dpc (*p* = 0.04) by ANOVA analysis. Both nasal and saliva fluid collection from ruminants are considered as non-invasive methods, whereas collection of OPF is invasive and difficult to do. Moreover, difficulty in collecting uniform OPF samples may contribute to inconsistent virus detection during consecutive probang sampling from known carrier animals. Cattle do not like and usually resist the collection of nasal fluids from their nostrils, whereas collection of saliva is much easier. Therefore, a salivary IgA test might be taken forward in outbreak situation in the field, even though the nasal IgA test has slightly superior test characteristics.

## Figures and Tables

**Figure 1 viruses-13-00814-f001:**
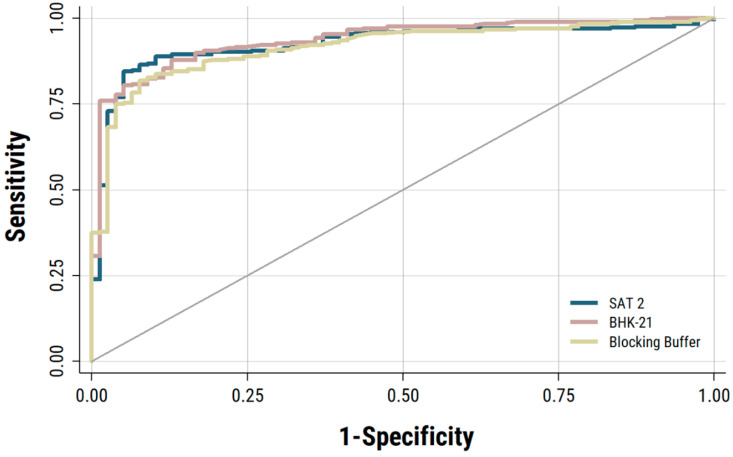
ROC curves estimated for IgA-ELISA using saliva samples collected from vaccinated un-infected and vaccinated-infected cattle. Three different negative antigen controls (FMDV SAT2, BHK-21 cell lysate, and a blocking buffer) were used to construct the IgA ELISA.

**Figure 2 viruses-13-00814-f002:**
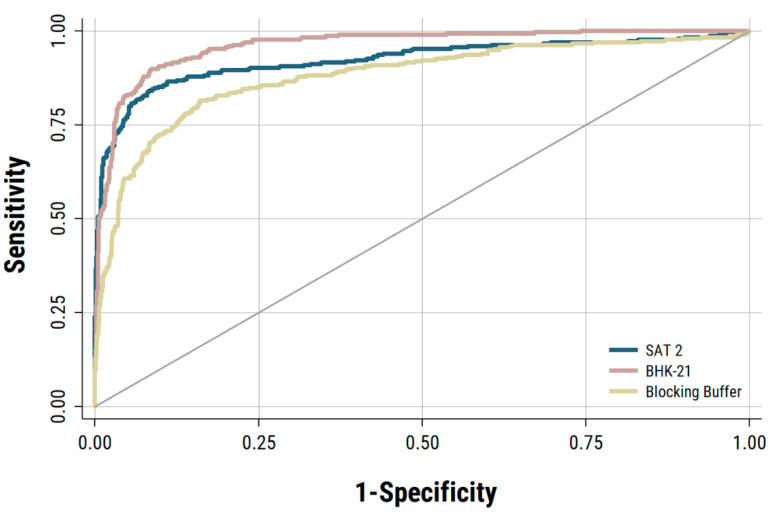
ROC curves estimated for IgA-ELISA using saliva samples collected from naïve un-infected and vaccinated-infected cattle. Three different negative antigen controls (FMDV SAT2, BHK-21 cell lysate, and a blocking buffer) were used to construct the IgA ELISA.

**Figure 3 viruses-13-00814-f003:**
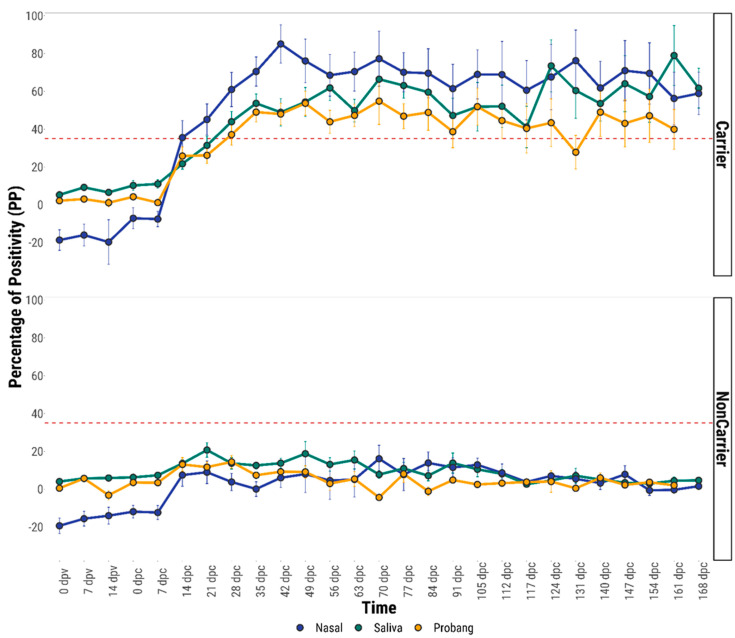
Comparative mean anti-FMDV IgA antibody responses in the mucosal (nasal, saliva and probang) fluids of vaccinated carrier and non-carrier cattle. The error bar represents the standard error. Red horizontal dashed lines indicate the diagnostic cut-off (35 PP).

**Figure 4 viruses-13-00814-f004:**
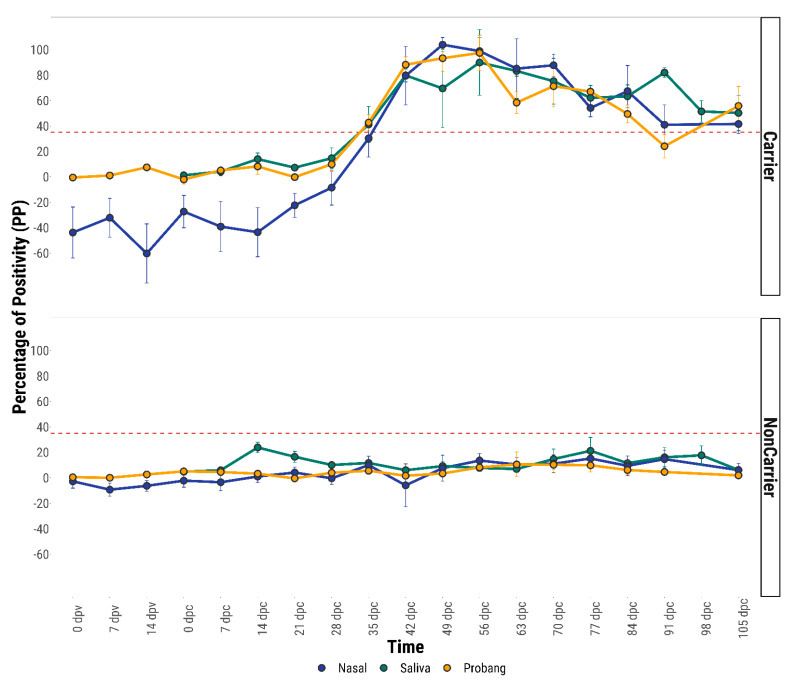
Comparative mean anti-FMDV IgA response in the mucosal (nasal, saliva and probang) fluids of unvaccinated carrier and non-carrier animals. The error bars represent the standard error. Red horizontal dashed lines indicate the diagnostic cut-off (35 PP).

**Figure 5 viruses-13-00814-f005:**
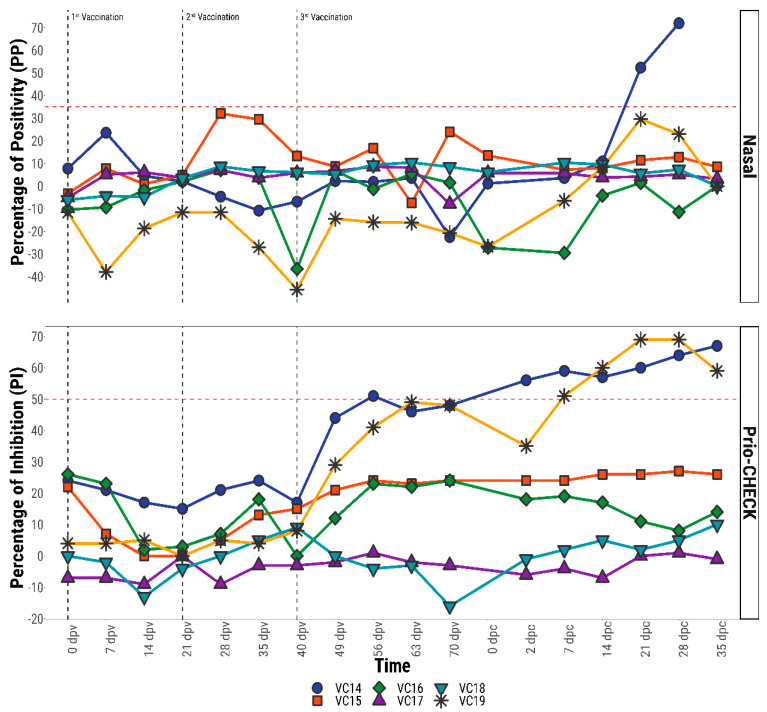
Antibody responses (FMDV specific IgA in nasal fluid and anti-NSP antibody in serum) after repeated vaccination in cattle. Vertical dashed black lines indicate the day of vaccination, whilst the horizontal dashed red lines indicate the diagnostic cut-offs for both the Nasal IgA-ELISA (35 PP) and the Prio-CHECK NSP ELISA (50 PI) tests. IgA antibody titre expressed as percentage of positivity (PP) values and anti-NSP antibody titre expressed as percentage of inhibition (PI) values.

**Table 1 viruses-13-00814-t001:** Diagnostic performance of IgA-ELISA estimated using saliva samples from vaccinated un-infected and vaccinated infected cattle. Diagnostic parameters are reported at different cut-off points using the three different negative antigen controls used to construct the IgA-ELISAs. PP = percentage of positivity; LR+ = likelihood ratio for a positive test; LR− = likelihood ratio for a negative test.

Negative Ag Control	PP	Sensitivity (Se)	Specificity (Sp)	LR+	LR−
*SAT 2*	20	84.46%	94.87%	16.470	0.164
	25	78.38%	84.87%	15.284	0.228
	30	73.65%	96.15%	19.149	0.274
	35	69.26%	97.44%	27.010	0.315
	40	65.54%	97.44%	25.561	0.354
*BHK-21*	20	77.10%	96.15%	20.047	0.238
	25	71.72%	98.72%	55.939	0.286
	30	66.33%	98.72%	51.727	0.341
	35	61.28%	98.72%	47.798	0.392
	40	57.91%	98.72%	45.172	0.426
*Blocking-Buffer*	20	73.40%	96.15%	19.084	0.277
	25	67.78%	97.44%	26.394	0.332
	30	62.96%	97.44%	24.556	0.380
	35	60.27%	97.44%	23.505	0.408
	40	54.55%	97.44%	21.273	0.466

**Table 2 viruses-13-00814-t002:** Diagnostic performance of IgA-ELISA estimated using saliva samples from naïve uninfected and vaccinated infected cattle. Diagnostic parameters are reported at different cut-off points using the three different negative antigen controls used to construct the IgA-ELISAs. PP = percentage of positivity; LR+ = likelihood ratio for a positive test; LR− = likelihood ratio for a negative test.

Negative Ag Control	PP	Sensitivity (Se)	Specificity (Sp)	LR+	LR−
*SAT 2*	20	84.12%	91.77%	10.039	0.173
	25	78.04%	94.82%	15.060	0.232
	30	73.65%	96.34%	19.647	0.274
	35	68.92%	97.60%	27.178	0.319
	40	65.54%	98.74%	49.538	0.349
*BHK-21*	20	77.10%	96.69%	23.311	0.237
	25	71.72%	97.02%	24.092	0.291
	30	66.33%	97.35%	25.067	0.346
	35	61.28%	97.79%	27.790	0.396
	40	57.91%	98.24%	32.829	0.428
*Blocking-Buffer*	20	73.40%	88.42%	6.340	0.301
	25	67.68%	92.06%	8.525	0.351
	30	62.96%	94.05%	10.575	0.394
	35	59.60%	95.59%	13.513	0.423
	40	54.55%	96.36%	14.992	0.472

**Table 3 viruses-13-00814-t003:** Diagnostic performance of the nasal IgA-ELISA. Diagnostic parameters are reported at different cut-off points. PP = percentage of positivity; LR+ = likelihood ratio for a positive test; LR− = likelihood ratio for a negative test. Diagnostic cut-off is set at 35PP.

PP	Se	Sp	LR+	LR−
10	90.48%	94.20%	15.589	0.101
20	84.35%	96.88%	26.993	0.161
30	78.23%	97.77%	35.048	0.223
35	76.53%	99.11%	85.714	0.237
40	74.49%	99.11%	83.428	0.255
45	69.73%	100%	-	0.303
50	65.65%	100%	-	0.343

**Table 4 viruses-13-00814-t004:** Diagnostic performance of the probang IgA-ELISA. Diagnostic parameters are reported at different cut-off points. PP = percentage of positivity; LR+ = likelihood ratio for a positive test; LR− = likelihood ratio for a negative test. Diagnostic cut-off is set at 35PP.

PP	Se	Sp	LR+	LR−
10	87.05%	93.65%	13.710	0.138
20	73.38%	97.88%	34.673	0.272
30	64.39%	98.94%	60.847	0.360
35	59.35%	99.47%	112.177	0.409
40	56.47%	99.47%	106.738	0.438
45	47.84%	99.47%	90.421	0.524
50	38.13%	99.47%	72.065	0.622

**Table 5 viruses-13-00814-t005:** Diagnostic parameters estimated for each of the IgA-ELISA by Bayesian modelling using probabilistic constraints. Sensitivity and specificity estimates are expressed as median (95% Bayesian Credible Interval).

	Sensitivity	Specificity
Saliva	0.79 (0.76–0.85)	0.99 (0.99–1.00)
Nasal	0.85 (0.80–0.93)	0.99 (0.99–1.00)
Probang	0.70 (0.64–0.78)	0.99 (0.99–1.00)

**Table 6 viruses-13-00814-t006:** Validation and comparison of nasal IgA-ELISA test efficiency with NSP assay (PrioCHECK-NSP) for detection of FMDV carriers. Assuming all carrier animals were detected by VI + RT-PCR assay (100%), all the other percentage values were calculated. VI = virus isolation; NSP = non-structural protein.

Animal Experiments	Clinically Infected/Vaccinated Challenged Animals	Vaccinated Carriers Detected by VI + RT-PCR	Carriers Detected by PrioCHECK-NSP Test	NSP Seroconversion by PrioCHECK-NSP Assay.	Carriers Detected by Nasal IgA-ELISA	Carriers Concordantly Detected by both PrioCHECK-NSP and IgA Test	Carriers Detected by Either PrioCHECK-NSP or IgA Test or Both
UV	0/20	9	7	10	8	7	9
UY	0/20	3	3	7	3	3	3
VH	5/20	9	9	18	9	9	9
VD/VE	6/20	11	10	12	9	8	10
TOT	11/80	32 (100%)	29 (90.62%)	47 (146.87%)	29 (90.62%)	27 (84.37%)	31 (96.87%)

**Table 7 viruses-13-00814-t007:** Comparative sensitivities of various virological and serological assays for detection of vaccinated and subsequently infected carrier cattle. Considering number of known carriers on different days of sampling period as 100%, sensitivities for all the other assays were calculated. Dpc = days post-challenge; VI = virus isolation; NSP = non-structural protein.

Sampling Days (dpc)	No. of Carriers	Carriers Detected by VI + RT-PCR	Carriers Detected by Nasal IgA	Carriers Detected by PrioCHECK-NSP	Carriers Detected by IgA+ PrioCHECK
28	32 (100%)	25 (78.1%)	24 (75%)	27 (84.37%)	30 (93.75%)
35	32 (100%)	25 (78.1%)	25 (78%)	28 (87.5%)	30 (93.75%)
42	23 (100%)	12 (52.17%)	18 (78.26%)	18 (78.26%)	20 (86.95%)
49	21 (100%)	14 (66.66%)	17 (80.95%)	16 (76.19%)	20 (95.23%)
55	21 (100%)	13 (61.90%)	16 (76.19%)	18 (85.71%)	19 (90.47%)
63	21 (100%)	7 (33.33%)	17 (80.95%)	18 (85.71%)	19 (90.47%)
77	21 (100%)	10 (47.61%)	16 (76.19%)	19 (90.47%)	19 (90.47%)
84	12 (100%)	4 (33.33%)	10 (83.33%)	10 (83.34%)	12 (100%)
91	12 (100%)	3 (25%)	8 (66.67%)	10 (83.34%)	11 (91.66%)
105	12 (100%)	2 (16.66%)	9 (75%)	9 (75%)	10 (83.34%)
112	9 (100%)	1 (11.11%)	7 (77.76%)	7 (77.76%)	7 (77.76%)
117	9 (100%)	1 (11.11%)	5 (55.56%)	7 (77.76%)	7 (77.76%)
124	9 (100%)	2 (22.22%)	6 (66.67%)	7 (77.76%)	8 (88.89%)
131	8 (100%)	1 (12.5%)	6 (75%)	6 (75%)	6 (75%)
140	8 (100%)	0 (0%)	6 (75%)	6 (75%)	6 (75%)
147	8 (100%)	2 (25%)	5 (62.5%)	6 (75%)	6 (75%)
154	8 (100%)	1 (12.5%)	6 (75%)	6 (75%)	6 (75%)
161	8 (100%)	2 (25%)	5 (62.5%)	6 (75%)	6 (75%)
168	8 (100%)	1(12.5%)	7 (87.5%)	6 (75%)	7 (87.5%)

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
