# Peer review of "Development and Validation of a Mucosal Antibody (IgA) Test to Identify Persistent Infection with Foot-and-Mouth Disease Virus"

_viruses, 2021, doi:10.3390/v13050814_

Round 1

Reviewer 1 Report

The manuscript by Biswal et al describe the Development and validation of a mucosal antibody (IgA) test to identify persistent infection with foot-and-mouth disease virus. This new diagnostic test based on specific anti FMDV secreted IgA for the detection of carrier animals using nasal, saliva and OPF samples could contribute to confirm or screen persistently infected cattle.

Data are well described and support the conclusions. I have only two specific comment and some minor comments.

Specific comments:

Authors describe the carrier state  in cattle as presence of virus in oro-pharyngeal or naso-pharyngeal cavity at 28 days or more days post-infection but recent publications reported on the divergence of this carrier state in catlle (carrier status could be defined as early as 10 days in vaccinated
cattle and 21 days in non-vaccinated cattle). What about the detection of such carrier cattle using the IgA -based developed in this study?

Abstract, Line 11: "Although transmission of FMDV from carrier cattle to naïve cattle has not been demonstrated experimentally": a bit misleading if you compare with lines 42-50 in the introduction section. The potential even low risk of transmission need to be stated in the abstract as well. Please correct.

Minor comments:

Abstract, Line 19: typo: "cab" => can, please correct

Introduction,Line 33: "within the picornaviridae family" => please correct into "within the  family picornaviridae"

Results,Line 194: "from the cmultiple vaccinated-challenged cattle" => typo, please correct

Line 315: "31 out of 32 vaccinated carries" => typo, please correct

Line 383: "sero-surveillence"=> orthographe: please correct

Author Response

Reviewer 1

Query-1 (Specific comments):

Authors describe the carrier state in cattle as presence of virus in oro-pharyngeal or naso-pharyngeal cavity at 28 days or more days post-infection but recent publications reported on the divergence of this carrier state in cattle (carrier status could be defined as early as 10 days in vaccinated cattle and 21 days in non-vaccinated cattle). What about the detection of such carrier cattle using the IgA -based developed in this study?

Answer 1:  Historically, it is known that up to 50% of FMD-recovered cattle may harbour virus in their oro-pharyngeal and naso-pharyngeal cavity at 28 or more days post-infection and are known as FMDV-carriers or persistently infected animals. However, as the reviewer pointed out a recent published report (Stenfeldt et al., 2016) described the divergence of this carrier state earlier in cattle- carrier status could be defined as early as 10 days in vaccinated cattle and 21 days in non-vaccinated cattle. However, the authors in Stenfeldt et al., 2016 study had used the Adeno-vectored FMDV vaccine for their study whereas in our study we have used conventional FMDV BEI inactivated vaccines. Further in their study they have used qPCR of oro-pharyngeal (OPF)fluid and we have done the qPCR of OPF as well as IgA ELISA using OPF, Saliva and Nasal fluids. In our study we described IgA is not detected in vaccinated cattle by the IgA ELISA using these three fluids whereas it started to be detected transiently in the acute phase of infection in unvaccinated animals and then continue to increase if the animals harbour virus in the Oro/Naso-pharynx and becomes a carrier in both unvaccinated and vaccinated carriers. Using Nasal fluid, IgA ELISA could detect many qPCR positive carriers by 14days post-challenge (dpc) and all the qPCR positive vaccinated carriers were detected by 21dpc (Fig 3). Using saliva samples in IgA ELISA all the vaccinated carriers were detected by 28dpc although many of them were detected on 21dpc. Similarly Using OPF IgA ELISA we could detect carrier animals by 28-35dpc. All these are shown in Figure 3 and described in the text in the section 3.2.3. Further unvaccinated carrier cattle were detected by 35dpc using all three IgA ELISA (Fig 4).

Query-2:  Abstract, Line 11: "Although transmission of FMDV from carrier cattle to naïve cattle has not been demonstrated experimentally": a bit misleading if you compare with lines 42-50 in the introduction section. The potential even low risk of transmission needs to be stated in the abstract as well. Please correct.

Answer 2:  As suggested by the reviewer suitable correction has been incorporated in the abstract.

Minor comments:

Query -3: Abstract, Line 19: typo: "cab" => can, please correct

Answer 3: Correction incorporated in the revised manuscript.

Query-4: Introduction, Line 33: "within the picornaviridae family" => please correct into "within the  family picornaviridae"

Answer 4: Correction incorporated in the revised manuscript.

Query-5: Results,Line 194: "from the cmultiple vaccinated-challenged cattle" => typo, please correct

Answer 5: Correction incorporated in the revised manuscript.

Query-6: Line 315: "31 out of 32 vaccinated carries" => typo, please correct

Answer 6: Correction incorporated in the revised manuscript.

Query-7: Line 383: "sero-surveillence"=> orthographe: please correct

Answer 7: Correction incorporated in the revised manuscript

Reviewer 2

Query 1: The authors should make an effort in correlating the levels of IgA with presence of virus in Probangs. For instance, it is known that probangs positivity is variable at the different time points. Could the authors detect significant lower levels of IgA when probangs were negative?

Answer 1: As mentioned by the reviewer the detection of FMDV virus/genome in the probang samples is variable at the different time points which is a probang technique issue. Therefore, a clear-cut consistent correlation between the levels of IgA antibody in the mucosal fluids with respect to the presence FMDV in the probang samples on that day is not expected and same has been seen in our IgA ELISAs. However, as soon as the animal loses the virus from oro/naso-pharynx, the level of anti-FMDV IgA antibody in IgA ELISAs reduced gradually and became undetected within couple of weeks. Please see further answers to query  number 5 below.

Query 2: Please, explain better the experimental design. It is difficult to follow as it stands. The authors first talk about 25 animals per experiment, but they only give specifics about 20. Are the other five controls? Also, there are animals challenged at 21 or 10 dpv but there is not specifics about what group is which.

Answer 2: In each cattle-challenge experiment 25 animals were used, and out of which 20 animals vaccinated and subsequently FMDV-challenged animals, while the 5-animals were used as un-vaccinated control animals for FMDV-challenge. Animals of the UV, UY series were challenged on 21 dpv, while animals of the VD/VE, and VH were vaccinated on 10 dpv. For better understanding of the animal-experimental design a supplementary table has been provided in the revised version of the manuscript explaining the details of the vaccine-challenge experiment and principal outcome of the animal experiments (Page 2, line 90-92 of the revised manuscript). Further we have included couple of sentences in the section 2.1.1.(line 74-92 paragraph).

Query 3: Figure 4 needs to be corrected. Please add carrier versus non-carriers. Also the orange color for probang samples does not match. If that is on purpose please explain.

Figure 5 needs lettering for each panel. Also, bottom panel X axis lettering appears cut. Also, there is supposed to have arrows indicating the different times of vaccination and only one arrow in the bottom panel can be seen.

Answer 3: Figures (3, 4 and 5) have been modified in the revised version of the manuscript.

Query 4: The authors in the discussion talk about increase of IgA during the acute phase of disease, however, these animals had been vaccinated and became carriers after subclinical disease. If they are talking about unvaccinated controls, Figure 4 only shows positive values over the threshols starting at 35dpc. Would the explain or correct the sentence?

Answer 4: Reviewer is right and we have corrected the sentence in the paragraph (line 390-402).

Query 5: How do the authors explain that in vaccinated animals once the levels of IgA have increased, they never decreased…did animals remained persistently infected? Would the authors considered collecting data for longer time until carrier state is cleared and compare with IgA levels?

Answer 5: The high level of anti-FMDV IgA response in the mucosal fluids could be due to the constant stimulation of local mucosal immune system by persistent FMDV on the dorsal soft palate or naso-pharynx. Therefore, an increased level of anti-FMDV IgA antibody could be detected as long as the vaccinated-infected animal remained persistently infected with FMDV. In our study, the carrier animals were kept in the animal containment facility up to 168 dpc and due to logistic demands the animal could not kept longer time until the clearance of their FMDV-carrier state. However, couple of carrier animals lost their status within this period and we could not recovered viruses in probang samples after consecutive sampling. In this couple of animals after one week of non-detection of virus by qPCR, we too could not detect IgA.

Minor changes:

Query 6: Line 19, “cab” should be ‘can’.

Answer 6: Correction has been included in the revised version of the manuscript.

Query 7: Line 194, ‘cmultiple’ should be ‘multiple’.

Answer 7: Correction has been included in the revised version of the manuscript

Query 8: There is not Figure 6a, but the text refers to it. Did the authors meant Fig 5a

Answer 8: Correction as figure 5a has been included in the revised version of the manuscript

Reviewer 2 Report

April 15th, 2021

Review: viruses-1185215-v1

“Development and validation of a mucosal antibody (IgA) test 2 to identify persistent infection with foot-and-mouth disease virus”.

In this manuscript the authors describe the development and standardization of an anti-IgA ELISA specific against foot-and-mouth disease virus as a method for screening of FMDV carrier animals. The topic is of great interest, since the carrier state represents a mayor backlash to gain FMD free status after an outbreak and it makes very difficult the development of eradication campaigns. Although the technique used is nothing innovative, the way the data are managed and analyzed clearly allow authors to make strong conclusions. The authors should make an effort in correlating the levels of IgA with presence of virus in Probangs. For instance, it is known that probangs positivity is variable at the different time points. Could the authors detect significant lower levels of IgA when probangs were negative?

Please, explain better the experimental design. It is difficult to follow as it stands. The authors first talk about 25 animals per experiment, but they only give specifics about 20. Are the other five controls? Also, there are animals challenged at 21 or 10 dpv but there is not specifics about what group is which.

Figure 4 needs to be corrected. Please add carrier versus non-carriers. Also the orange color for probang samples does not match. If that is on purpose please explain.

Figure 5 needs lettering for each panel. Also, bottom panel X axis lettering appears cut. Also, there is supposed to have arrows indicating the different times of vaccination and only one arrow in the bottom panel can be seen.

The authors in the discussion talk about increase of IgA during the acute phase of disease, however, these animals had been vaccinated and became carriers after subclinical disease. If they are talking about unvaccinated controls, Figure 4 only shows positive values over the threshols starting at 35dpc. Would the explain or correct the sentence?

How do the authors explain that in vaccinated animals once the levels of IgA have increased, they never decreased…did animals remained persistently infected? Would the authors considered collecting data for longer time until carrier state is cleared and compare with IgA levels?

Minor changes:

  1. Line 19, “cab” should be ‘can’.
  2. Line 194, ‘cmultiple’ should be ‘multiple’.
  3. There is not Figure 6a, but the text refers to it. Did the authors meant Fig 5a

Author Response

Reviewer 2

Query 1: The authors should make an effort in correlating the levels of IgA with presence of virus in Probangs. For instance, it is known that probangs positivity is variable at the different time points. Could the authors detect significant lower levels of IgA when probangs were negative?

Answer 1: As mentioned by the reviewer the detection of FMDV virus/genome in the probang samples is variable at the different time points which is a probang technique issue. Therefore, a clear-cut consistent correlation between the levels of IgA antibody in the mucosal fluids with respect to the presence FMDV in the probang samples on that day is not expected and same has been seen in our IgA ELISAs. However, as soon as the animal loses the virus from oro/naso-pharynx, the level of anti-FMDV IgA antibody in IgA ELISAs reduced gradually and became undetected within couple of weeks. Please see further answers to query  number 5 below.

Query 2: Please, explain better the experimental design. It is difficult to follow as it stands. The authors first talk about 25 animals per experiment, but they only give specifics about 20. Are the other five controls? Also, there are animals challenged at 21 or 10 dpv but there is not specifics about what group is which.

Answer 2: In each cattle-challenge experiment 25 animals were used, and out of which 20 animals vaccinated and subsequently FMDV-challenged animals, while the 5-animals were used as un-vaccinated control animals for FMDV-challenge. Animals of the UV, UY series were challenged on 21 dpv, while animals of the VD/VE, and VH were vaccinated on 10 dpv. For better understanding of the animal-experimental design a supplementary table has been provided in the revised version of the manuscript explaining the details of the vaccine-challenge experiment and principal outcome of the animal experiments (Page 2, line 90-92 of the revised manuscript). Further we have included couple of sentences in the section 2.1.1.(line 74-92 paragraph).

Query 3: Figure 4 needs to be corrected. Please add carrier versus non-carriers. Also the orange color for probang samples does not match. If that is on purpose please explain.

Figure 5 needs lettering for each panel. Also, bottom panel X axis lettering appears cut. Also, there is supposed to have arrows indicating the different times of vaccination and only one arrow in the bottom panel can be seen.

Answer 3: Figures (3, 4 and 5) have been modified in the revised version of the manuscript.

Query 4: The authors in the discussion talk about increase of IgA during the acute phase of disease, however, these animals had been vaccinated and became carriers after subclinical disease. If they are talking about unvaccinated controls, Figure 4 only shows positive values over the threshols starting at 35dpc. Would the explain or correct the sentence?

Answer 4: Reviewer is right and we have corrected the sentence in the paragraph (line 390-402).

Query 5: How do the authors explain that in vaccinated animals once the levels of IgA have increased, they never decreased…did animals remained persistently infected? Would the authors considered collecting data for longer time until carrier state is cleared and compare with IgA levels?

Answer 5: The high level of anti-FMDV IgA response in the mucosal fluids could be due to the constant stimulation of local mucosal immune system by persistent FMDV on the dorsal soft palate or naso-pharynx. Therefore, an increased level of anti-FMDV IgA antibody could be detected as long as the vaccinated-infected animal remained persistently infected with FMDV. In our study, the carrier animals were kept in the animal containment facility up to 168 dpc and due to logistic demands the animal could not kept longer time until the clearance of their FMDV-carrier state. However, couple of carrier animals lost their status within this period and we could not recovered viruses in probang samples after consecutive sampling. In this couple of animals after one week of non-detection of virus by qPCR, we too could not detect IgA.

Minor changes:

Query 6: Line 19, “cab” should be ‘can’.

Answer 6: Correction has been included in the revised version of the manuscript.

Query 7: Line 194, ‘cmultiple’ should be ‘multiple’.

Answer 7: Correction has been included in the revised version of the manuscript

Query 8: There is not Figure 6a, but the text refers to it. Did the authors meant Fig 5a

Answer 8: Correction as figure 5a has been included in the revised version of the manuscript